# Gender Differences in the Impact of Recommendations on Diagnostic Imaging Tests: A Retrospective Study 2007–2021

**DOI:** 10.3390/life13020289

**Published:** 2023-01-20

**Authors:** Lucy A. Parker, Andrea Moreno-Garijo, Elisa Chilet-Rosell, Fermina Lorente, Blanca Lumbreras

**Affiliations:** 1Department of Public Health, University Miguel Hernández de Elche, 03550 Alicante, Spain; 2CIBER de Epidemiología y Salud Pública (CIBERESP), 28029 Madrid, Spain; 3Faculty of Pharmacy, University Miguel Hernández de Elche, 03550 Alicante, Spain; 4Radiology Department, University Hospital of San Juan de Alicante, Sant Joan d’Alacant, 03550 Alicante, Spain

**Keywords:** imaging tests, radiation exposure, recommendations, gender, socioeconomical status

## Abstract

(1) Background: The frequency of imaging tests grew exponentially in recent years. This increase may differ according to a patient’s sex, age, or socioeconomic status. We aim to analyze the impact of the Council Directive 2013/59/Euratom to control exposure to radiation for men and women and explore the impact of patients’ age and socioeconomic status; (2) Methods: The retrospective observational study that includes a catchment population of 234,424. We included data of CT, mammography, radiography (conventional radiography and fluoroscopy) and nuclear medicine between 2007–2021. We estimated the associated radiation effective dose per test according using previously published evidence. We calculated a deprivation index according to the postcode of their residence. We divided the study in 2007–2013, 2014–2019 and 2020–2021 (the pandemic period). (3) Results: There was an increase in the number of imaging tests received by men and women after 2013 (*p* < 0.001), and this increase was higher in women than in men. The frequency of imaging tests decreased during the pandemic period (2020–2021), but the frequency of CT and nuclear medicine tests increased even during these years (*p* < 0.001) and thus, the overall effective mean dose. Women and men living in the least deprived areas had a higher frequency of imaging test than those living in the most deprived areas. (4) Conclusions: The largest increase in the number of imaging tests is due to CTs, which account for the higher amount of effective dose. The difference in the increase of imaging tests carried out in men and women and according to the socioeconomic status could reflect different management strategies and barriers to access in clinical practice. Given the low impact of the available recommendations on the population exposure to radiation and the performance of high-dose procedures such as CT, deserve special attention when it comes to justification and optimization, especially in women.

## 1. Introduction

In recent decades, diagnostic imaging tests using radiation, together with nuclear medicine, have been a major source of exposure to non-natural radiation in the general population in Western countries [1]. In most developed countries, it has been shown that the contribution of nuclear medicine diagnostic procedures is between 4% and 14% [2]. The widespread use of diagnostic techniques such as CT scans has also meant the number of doses of ionising radiation received has increased, at both individual and population level [3]. Over the last 22 years, radiation due to CT exposures were estimated to account for 0.7% of the cancer incidence and 1% of cancer mortality [4]. Moreover, the incidence of cancer in individuals who had been exposed to CT was found to be 24% higher compared to individuals who had not [4]. Other adverse effects of the increase in imaging tests are time and resource utilization and the presence of incidental findings that can lead to unnecessary clinical interventions [5].

The US Food and Drug Administration (FDA) and the European Union have developed strategies aimed at reducing unnecessary radiation exposure. In 2010, the FDA published an initiative to promote patient safety through the justification of each imaging test carried out in accordance with the patient’s symptoms and medical history. In addition, they promoted the lowest possible dose based on each patient’s anatomical and physiological factors [6]. The most relevant European initiative in this regard is the Directive 97/432/EURATOM [7] which has given rise to various projects such as the DOSE DATAMED. This project consisted of a survey to assess the radiation received by the European population. In accordance with this European directive other projects have been developed, which aim at unifying the training and knowledge of the professionals involved in radiation protection: MEDRAPET Project [8], European Medical ALARA Network [9] and Medical Physics Expert Project [10]. The Directive 2013/59/Euratom [11], which was an update of the previous directive, was published in 2013. It was supposed to be transposed in all member countries before the 6 February 2018. However, in Spain, this directive was not transposed until the 18 October 2019 (Royal Decree 601/2019 [12]) and has not yet been implemented. 

Despite these recommendations, recent studies indicate that the frequency of imaging tests has grown exponentially in recent years. Even during the SARS-CoV-2 pandemic when access to healthcare was limited, the use of imaging tests remained widespread [13]. 

Moreover, this increase in the frequency of imaging tests may differ according to patient’s characteristics such as sex, age, or socioeconomic status. It has been known for over 30 years that there are wide differences in the clinical management of men and women in many situations. Even though women use more health services than men [14], there is evidence that there is a diagnostic bias between men and women [15]. Research regarding appropriateness has shown that women are less likely to have an imaging test considered adequate than men [16]. Another study also found that more inappropriate and uncertain myocardial perfusion imaging was ordered for women compared to men [17]. Some studies estimate that radiation from imaging tests may be linked to 1% of cancers diagnosed in the United States, with adult women between the ages of 35 and 54 years being the largest population at risk [18,19] especially if they are under the age of 30 years [20,21]. Previous reports identified ionising radiation from CT as a contributing factor for both breast cancer [22] and a greater hazard of radiation-related solid cancer in women compared to men [23]. These data could have influenced the differential use of imaging diagnostic testing in women compared to men [24]. 

In addition, other aspects such as the socioeconomic context could also play a role in the different performance of imaging tests. Social determinants of health are known to play a large role in health outcomes. For example, they may impact an individual’s ability to access nutritious food and healthcare resources, time, and space for physical activity [25,26]. A previous study found that the intersection of sex and social factors in influencing patient-relevant outcomes varies even among countries with similar healthcare and high gender equality [27]. High income countries have been associated with a higher frequency of CT examinations per inhabitant [28]. However, there is little data on the differences in the frequency of imaging tests according to the socioeconomic status of the population for men and women in the same country, including those with a public health system like Spain. 

The purpose of this study was to analyze the impact of the establishment of new recommendations in 2013 to control exposure to radiation for men and women, and to explore the impact of patients’ age and socioeconomic status on gender differences, in a single university hospital. 

## 2. Materials and Methods

### 2.1. Study Design

We conducted a retrospective observational study to analyse the impact of the established recommendations in 2013 on tests performed in clinical practice according in men and women. We also calculated the per capita effective dose and the influence of patients’ age and socioeconomic status.

### 2.2. Setting

The target population for the study were all residents in the catchment area of San Juan Hospital (Alicante), in the Valencian Community (Spain), a general centre, with an estimated catchment population of 234,424. This is a referral hospital for all individuals living in the catchment area who belong to the National Health Care System (NHS). Most of the Spanish population uses the NHS as the main medical service (the publicly funded insurance scheme covers 98.5% of the Spanish population).

### 2.3. Participants

We included utilization data of CT, radiography (including mammography, conventional radiography, and fluoroscopy) and nuclear medicine by the target population between 2007–2021 (in any care setting, inpatient, outpatient, or emergency department). We excluded imaging tests that did not involve radiation exposure (i.e., MRI and ultrasound) and patients who had an imaging test in this hospital but did not belong to its catchment area.

### 2.4. Imaging Test Frequency

For collecting data on imaging test frequency, we carried out procedures similar to those used in a previous study [29]. Briefly, we collected the following data from the Medical Image Bank of the Valencian Community from the Department of Universal Health and Public Health Service: sex and age at entry in the study, radiological examination, and date. Both the images and the patient data were anonymised and deidentified by the Health Informatics Department of the Hospital of San Juan using Research and Development (R&D) Cloud CEIB Architecture [30]. This digital register started in 2007 in our setting. Each imaging test received was classified as a single radiation exposure. However, abdomen and pelvis tests carried out in the same process were included as a single abdomen–pelvis test, while an abdomen or a pelvis test in a different process, even in the same patient, were included as two different tests. Thoracic and lumbar spine tests were included when they were performed alone but not when performed together with chest or abdominal tests.

### 2.5. Effective Dose Estimate

Given that it was impossible to get individual machine parameters for all imaging tests, we estimated the associated radiation effective dose per test according to its region of anatomical coverage by age and using previously published evidence [31]. This review provides values of the typical effective doses associated with the 20 most frequent imaging tests for adults and children and for the most widely used set of weights (ICRP60) as well as for the most recent (ICRP103). In addition, we estimated the effective dose of imaging tests different from the 20 most frequent imaging tests in Dose DataMed 2 project according to previous studies [32,33,34]. 

### 2.6. Socioeconomical Status

To represent the socioeconomic status of the individuals we calculated a deprivation index according to the postcode of their residence. The Spanish Society of Epidemiology (SEE) published a deprivation index for the entire country using the enumeration district [35] and the information from the 2011 census. We reconstructed the index within the catchment area of the hospital by assigning each enumeration district with the postcode and then estimated the six socioeconomic indicators used by the SEE at postcode level using the census data (percentage of manual working population, percentage of casual working population, percentage of unemployed population, percentage of population with insufficient education, percentage of young population with insufficient education, and percentage of main dwellings without internet access). We used principal components analysis in Stata SE to recalculate the deprivation index by postcode. As this is a standardized index (with mean 0 and standard deviation 1), values close to zero would indicate the average deprivation in the population area. We divided the population in tertials according to the deprivation index: least deprived between −2.579506 and −1.060423, medium deprived between −0.813536 and 0.276083, and most deprived between 0.399323 and 5.022990.

### 2.7. Calendar Time

We assessed the different values according to two periods of study: 2007–2013 and 2014–2019 according to the year of publication of the Council Directive 2013/59/Euratom. Although this recommendation was transposed in Spain on the 18 October 2019, we did not evaluate its impact because of the SARS-CoV-2 pandemic in 2020. However, we also assessed the pandemic period: 2020–2021.

### 2.8. Statistical Analysis

All analysis were stratified by sex. We evaluate the frequency of imaging tests performed by imaging modality, age (<18 years [children]; 18–64 years [adults], and >64 years [older adults]), deprivation index (grouped in terciles), and calendar year (2007–2013, 2014–2019, 2020–2021) using the Chi-Square test. The annual average frequency was assessed per 1000 persons per women and men (number of people who are administratively assigned to the university hospital in each year by sex and age group). We also estimated the effective radiation dose by imaging modality (median and interquartile range) age, deprivation, and calendar time using Mann–Whitney U test. 

Statistical analyses of the data were performed with SPSS (V.25.0; SPSS). A *p*-value of 0.05 was considered significant.

## 3. Results

### 3.1. Population Included in the Study

In 2007, 232,446 people were administratively assigned to the selected hospital: 107,622 (46.3%) men and 124,824 (53.7%) women. The population in 2021 was 249,572 persons: 114,434 (45.9%) men and 135,138 (54.1%) women. There were not statistical differences according to the distribution of demographic variables (age and deprivation level) between men and women in the years 2007–2021: 16% were subjects < 18 years, 61% were subjects 18–64 years, and 23% >65 years. The deprivation level was divided in terciles, with the minimum value −2.84 (least deprived) and the maximum 5022 (mean −0.9514, sd 2.01).

### 3.2. Impact of Calendar Time in the Frequency of Imaging Tests According to Type of Imaging Test, Patients’ Age and Deprivation Index for Men and Women (Table 1)

The frequency of imaging tests was higher in women than in men for the three periods of study. The increase in tests between the years 2007–2013 and 2014–2019 was higher in women than in men (from 575.3 tests per 1000 women to 634.3 tests per 1000 women, percentage of change of 10.3% vs. from 498.5 tests per 1000 men to 535.6 tests per 1000 men, percentage of change of 7.4%). The frequency of imaging tests decreased in 2020–2021 to 519.6 tests per 1000 women (percentage of change of −18.1%) (*p* < 0.001) and this decrease was lower in men during the years 2020–2021, 471 tests per 1000 men (percentage of change −12.1%) (*p* < 0.001). (Table 1).

Although the frequency of radiography in men was similar between 2007–2013 and 2014–2019, it decreased during the years 2020–2021. CT frequency in men, in contrast, increased throughout the period of study (67.1 tests per 1000 persons in 2007–2013, 88.8 tests per 1000 persons in 2014–2019, and 98.7 tests per 1000 persons in 2020–2021). Similarly, the frequency of nuclear medicine tests in men also increased during the period of study: 29.9 per 1000 men in 2007–2013, 38.4 per 1000 men in 2014–2019, and 41.2 per 1000 men in 2020–2021 (*p* < 0.001). In women, the frequency of mammography, radiography and nuclear medicine increased between 2007–2013 (26.3 per 1000 women, 460.4 per 1000 women, and 35.6 per 1000 women, respectively) and 2014–2019 (32.5 per 1000 women, 486.7 per 1000 women, and 43.4 per 1000 women, respectively) and these frequencies decreased in 2020–2021 (28.7 per 1000 women, 376.3 per 1000 women, and 37.7 per 1000 women, respectively). However, the frequency of CT also increased during the study (*p* < 0.001).

According to age, the frequency of tests decreased in 2014–2019 and 2020–2021 in comparison with 2007–2013 for men < 18 years and those aged 18–64 years; the frequency increased in men > 64 years between 2007–2013 and 2014–2019 (686.2 tests per 1000 men to 892.3 tests per 1000 men) and this frequency decreased in 2020–2021 (855.8 tests per 1000 men) (*p* < 0.001). Although the frequency of imaging tests in women < 18 years was lower than in men during the three periods of study, the frequency of tests in women > 64 years was higher than in men > 64 years during the period of study. As well as in men, the frequency of tests also decreased in women < 18 years and those 18–64 years in 2014–2019 and in 2020–2021 in comparison with 2007–2013; nevertheless, the frequency of imaging tests increased in women > 64 years in 2014–2019 in comparison with 2007–2013 (916.6 tests per 1000 women to 1180 tests per 1000 women) and this frequency decreased in 2020–2021 (1010.3 tests per 1000 women) in comparison with 2014–2019 (*p* < 0.001) 

In Figure 1, we have shown that the amount of CT increased in 2014–2019 and in 2020–2021 in comparison with 2007–2013 in patients aged 18–64 years and in those older than 64 years, for men and women. The frequency of radiographies increased in 2014–2019 in comparison with 2007–2013 in patients > 64 years, although it decreased in patients < 18 years and in those aged 18–64 years in men and women. In addition, although the frequency of mammographies decreased during 2020–2021 in women aged 18–64 years, it increased in women older than 64 years. The frequency of nuclear medicine tests increased between 2007–2013 and 2014–2019 in men and women > 64 years and it decreased in 2020–2021.

**Table 1 life-13-00289-t001:** Description of the impact of calendar time in the frequency of imaging tests according to type of imaging test; patients’ age and deprivation index for men and women.

Frequency per 1000 Persons	Men				Women			
	2007–2013	2014–2019	2020–2021	*p* Value	2007–2013	2014–2019	2020–2021	*p* Value
Imaging test				<0.001				<0.001
Mamography	0.9	0.9	0.8		26.3	32.5	28.7	
Radiography	400.5	407.6	330.4		460.4	486.7	376.3	
CT	67.1	88.8	98.7		53.0	71.7	76.9	
Nuclear medicine	29.9	38.4	41.2		35.6	43.4	37.7	
Age (years)				<0.001				<0.001
<18	409.4	327.3	223.4		310.5	255.6	168.1	
18–64	452.2	458.4	393.9		519.4	532.6	431	
>64	686.2	892.3	855.8		916.6	1180	1010.3	
Deprivation index				<0.001				<0.001
Least deprived	414.5	474.4	440.1		492.1	587.1	500.9	
Medium deprived	407.6	466.1	422.1		452.3	537.0	456.6	
Most deprived	323.1	346.1	310.4		400.1	448.5	376.2	
Total	498.5	535.6	471.0	<0.001	575.3	634.3	519.6	<0.001

The frequency of imaging tests was higher in those patients living in the least deprived areas than in those living in the most deprived areas for women and men (Figure 2). In men and women, the frequency of imaging tests increased between the years 2007–2013 and 2014–2019 for the three groups of patients according to their living area and decreased in the years 2020–2021 (*p* < 0.001). In Figure 2, the frequency of radiographies decreased for men and women in 2020–2021 in comparison with 2014–2019 and with 2007–2013 regardless the deprivation area; however, the amount of CT increased in the three periods of time for men and women in the three deprivation groups. In women, the frequency of mammographies decreased in 2020–2021 in comparison with 2014–2019, but it was higher than the frequency in 2007–2013. In men, the frequency of nuclear medicine tests increased in 2020–2021 in comparison with the previous years, mainly in men in the least deprived area; in women, the frequency of nuclear medicine tests decreased in 2020–2021 in comparison with the previous years for women regardless of the deprivation index.

The impact of calendar time in the mean dose (mSv), according to the type of imaging test, patients’ age and deprivation index for men and women, is shown in Table 2.

Men received higher effective doses of radiation than women in the three periods of time, and there was an increase in the effective mean dose received throughout the period of study for men and women.

In men, the mean dose associated with radiographs increased between 2007–2013 and 2014–2019 (0.14 mSv and 0.20 mSv, respectively), and decreased in 2020–2021 (0.17 mSv). This trend was the same for nuclear medicine tests: the frequency increased between 2007–2013 and 2014–2019 (0.32 mSv and 0.36, respectively), and decreased in 2020–2021 (0.30 mSv). In contrast, the mean dose associated with CT increased throughout the period of study (*p* < 0.001). The same trend was found in women throughout the period of study (*p* < 0.001).

Regarding age, the mean dose increased in men > 64 years throughout the study (2007–2013, 2.20 mSv; 2014–2019, 3.18 mSv, and 2020–2021, 5.59 mSv) (percentage of change between 2007–2013 and 2020–2021, 154%%) (*p* < 0.001). The mean dose received in women > 64 years also increased throughout the period of study and the percentage of change between 2007–2013 and 2020–2021 was lower than in men (45%) (*p* < 0.001). 

Men living in the least deprived areas received the higher mean dose in comparison with men living in the medium and the most deprived areas. In addition, the mean dose received by men living in the least deprived areas increased 59.7% between 2007–2013 (0.77 mSv) and 2020–2021 (1.23 mSv). In men living in the most deprived areas, the mean dose received increased by 30.4% between 2007–2013 (0.60 mSv) and 2020–2021 (0.90mSv) (*p* < 0.001). Women living in the least deprived areas also showed the higher mean dose in comparison with women living in the medium and the most deprived areas. The mean dose received by women living in the least deprived areas increased by 45.7% between 2007–2013 (0.70 mSv) and 2020–2021 (1.02 mSv). In women living in the most deprived area, the mean dose received increased by 28.1% between 2007–2013 (0.64 mSv) and 2020–2021 (0.82 mSv) (*p* < 0.001).

## 4. Discussion

The results provide important information on the imaging diagnostic test trend and the collective effective dose, according to the available recommendations in a university hospital for men and women. We found an increase in the number of imaging tests received by men and women after the publication of the recommendations in 2013, and this increase was higher in women than in men. The frequency of imaging tests, including all imaging modalities, decreased during the pandemic period (2020–2021), but the frequency of CT increased even during these years in men and women, and the frequency of nuclear medicine tests in men and thus, the overall effective mean dose. The population aged > 64 years showed the highest frequency of imaging tests and the frequency increased between 2007–2013 and 2014–2019 for both women and men. Moreover, women and men living in the least deprived areas had a higher frequency of imaging test (and the higher mean dose received) than those living in the most deprived areas.

The publication of several recommendations to decrease the population’s exposure to radiation did not have an impact on the imaging tests that were carried out. A previous study on clinicians’ awareness of the European recommendations showed that nearly 80% of the clinicians surveyed had never heard of the European recommendations [36]. In addition, they did not take into account the need to consider radiation exposure when ordering imaging tests and the requirement to inform the patient about the risks associated with medical radiation exposure [36]. Women showed a greater increase in both the number of CT scans and conventional radiographies received than men after the publication of the recommendations, especially among patients older than 64 years. 

The frequency of nuclear medicine diagnostic procedures increased in 2014–2019 in comparison with 2007–2013 and it also increased in 2020–2021 in men. Nuclear medicine examinations contributed approximately 6% to the total frequency of all diagnostic medical procedures included in this study, leading to a high exposure to radiation. Similar data have been shown in other countries which followed DoseDataMed2 methodology [37], and therefore, it is essential to raise awareness about the potential dose optimization. Although previous studies found a remarkable increase in the number of CTs performed annually [38,39,40], they also found that CT utilization plateaued due to several reasons such as the acknowledgement of the harms of excess imaging, an increase in the use of other modes of imaging, or the cost [41]. However, we found that the number of CTs increased during the period of study regardless of the publication of recommendations and the pandemic period. During the pandemic in 2020–2021, the lower number of imaging tests reflected the much smaller number of patients treated that year. However, as previous research showed [13], the COVID-19 lung imaging recommendations which covered all radiological modalities, in particular the CT, could reflect the further increase in the number of CTs in 2020–2021. Moreover, the reason for the increase in CTs during the period of study could be that modern helical CT enables faster examination; thus, more examinations can be performed in one day [42]. The decreasing number of conventional radiographies is also probably due to the tendency of replacing it with CT [28]. CTs only represent 13% of all radiological procedures in our setting, but they are the major source of exposure to the population and recent reports have shed light on the increasing frequency of CT. For example, a recent study comprising 2.5 million patients found that patients underwent a median of six CT exams in a year and that some patients received up to 109 exams over five years [43]. Thus, more efforts are needed to increase optimisation and justification according to established recommendations. 

The number of CTs carried out was higher in men than in women. In contrast, the number of conventional radiographies was higher in women than in men, mainly due to breast cancer examinations. Nevertheless, the percentage of change between 2007–2013 and 2014–2019 for both conventional radiographies and CTs was higher in women than in men. In addition, the number of nuclear medicine diagnostic procedures was higher in women than in men in 2007–2013 and 2014–2019; however, in the pandemic years the frequency increased in men and decreased in women. This could reflect different clinical management strategies in practice for women and men. A previous study on the clinical management of solitary pulmonary nodule found that women were more likely than men to have a follow-up rather than have an immediate intervention [44]. As a result, accumulative radiation was higher in women than in men. It is necessary to include sex as a variable in future lines of research, as well as in protocols for action in relation to imaging tests and their indications in clinical guidelines since, despite the evidence on the differences between men and women, the incorporation of sex as a variable is not very common at present. Reviews of clinical guidelines have shown that only 35% of them take sex into account as a specific factor for the detection, diagnosis, or management of different diseases in clinical settings [45].

The healthcare system in Spain is public; that is, it is free of charge to anyone living and working in Spain, and the general taxation funds the Spanish state healthcare system. However, patients living in the least deprived areas had a higher frequency of imaging tests and received a higher effective dose than those living in the most deprived areas. On the one hand, if we consider the risk of excessive radiation, this could be interpreted with an equity lens in a positive manner (i.e., lower radiation among the most vulnerable population). However, it would be important to explore the pertinence of the imaging tests solicited and ensure that this reduction in imaging among the most deprived population is not due to a reduction of the pertinent tests due to problems with access, as this could lead to diagnostic delays.

We included a general hospital and its catchment area (with a total population of over 200,000 people). Even though our results could have some limited generalizability in other settings, the population included in this study is similar to the general Spanish population. Moreover, analysing this population provides important insights, showing as far as we know, the first evaluation of the impact of the available recommendations on the frequency of imaging tests carried out for men and women. Limitations of the study need to be included. This study did not consider images carried out in private health and it may hide differences by socioeconomical status. In addition, the deprivation index is a population level type indicator. Its advantage is that it represents a summary measure of the socio-economic characteristics of the population residents in each census section, which allows the study of socio-economic inequalities in health. It can be considered a measure of socio-economic deprivation of the census section, combining information related to individuals (compositional) and to the context. Although this method has the advantage of a similar population size dimension, it is sensitive to variation and it could change over time and given that, it includes 21 different areas. We obtained the data from the Medical Image Bank of the Valencian Community from the Department of Universal Health and Public Health Service and did not distinguish between the slice-spiral CT and the multi-slice CT. However, according to a previous study [46], the average effective dose to patients was only slightly changed from 7.4 mSv at single-slice to 5.5 mSv and 8.1 mSv at dual- and quad-slice scanners, respectively. In our study, we applied an estimation of the average values of the effective dose of imaging tests, and thus, we do not consider that this variation according to the type of CT could have influenced on the results.

## 5. Conclusions

In conclusion, this is the first study to provide insight into the impact of available recommendations on population exposure due to radiological medical procedures for men and women during a long period of study. By far, the largest increase in the number of imaging tests is due to CTs and nuclear medicine tests, which account for the higher amount of the effective dose. The difference in the increase of imaging tests carried out in men and women and according to the socioeconomic status could reflect different management strategies in clinical practice. Given the low impact of the available recommendations on the population’s exposure to radiation, the performance of high-dose procedures, such as CT and nuclear medicine tests, deserve special attention when it comes to justification and optimization, especially in women.

## Figures and Tables

**Figure 1 life-13-00289-f001:**
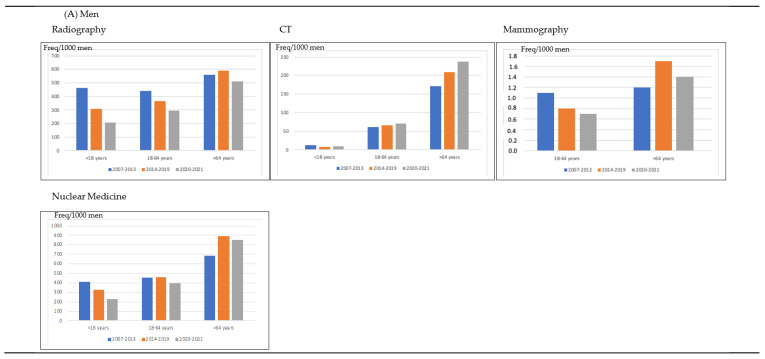
Distribution of the frequency of imaging test per 1000 habitants per men and women according to the patients’ age and calendar time.

**Figure 2 life-13-00289-f002:**
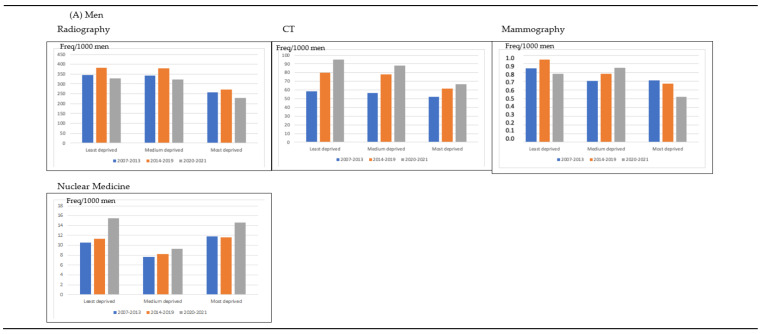
Distribution of the frequency of imaging test per 1000 habitants per men and women according to the patients’ deprivation index and calendar time.

**Table 2 life-13-00289-t002:** Description of the impact of calendar time in the mean dose (mSv) according to the type of imaging test; patients’ age and deprivation index for men and women.

Mean Dose (mSv)	Men				Women			
	2007–2013	2014–2019	2020–2021	*p* Value	2007–2013	2014–2019	2020–2021	*p* Value
Imaging test				<0.001				<0.001
Mamography	0	0	0		0.01	0.01	0.01	
Radiography	0.14	0.20	0.17		0.17	0.24	0.20	
CT	0.57	0.71	0.80		0.44	0.56	0.63	
Nuclear medicine	0.32	0.36	0.30		0.37	0.43	0.32	
Age				<0.001				<0.001
<18	0.31	0.16	0.22		0.25	0.21	0.15	
18–64	0.80	1.00	1.05		0.83	1.04	1.05	
>64	2.20	3.18	5.59		1.98	2.80	2.88	
Deprivation index				<0.001				<0.001
Least deprived	0.77	1.02	1.23		0.70	0.97	1.02	
Medium deprived	0.71	0.96	1.07		0.63	0.85	0.91	
Most deprived	0.69	0.82	0.90		0.64	0.78	0.82	
Total	1.04	1.27	1.27	<0.001	0.99	1.24	1.16	<0.001

## Data Availability

The data presented in this study are available on request from the corresponding author.

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
