# Peer review of "Gender Differences in the Impact of Recommendations on Diagnostic Imaging Tests: A Retrospective Study 2007–2021"

_life, 2023, doi:10.3390/life13020289_

Round 1

Reviewer 1 Report

The authors presented a retrospective study on the impact of recommendations to control exposure to radiation on diagnostic imaging frequency and dose with gender, age and socioeconomic status as factors. The study is properly designed and the conclusion revealed that there is a need to further improve the awareness and implementation of the recommendations and to pay additional attention regarding justification of diagnostic imaging in clinical practice. Overall, the article is organized and well-written. I would suggest publishing after the following comments are addressed.

  1. In Line 33, the last sentence in the abstract is not finished.

  2. In Line 40-42, reference should be added for the statement: “Over the last 22 years, radiation due to CT exposures were estimated to account for 0.7% of the cancer incidence and 1% of cancer mortality”.

  3. In Line 173, can the author check on the minimum deprivation index value? Is it -284 or -2.84? In addition, details on the reconstruction method of deprivation index and the deprivation index ranges for the three deprivation levels should be provided in the article.

  4. There are multiple places in the article, where “,” and “.” are misused in numbers, making it difficult to comprehend, such as “750.9 tests per 1.000 men” in Line 204. The authors should go through the manuscript and fix all the misuse.

  5. In Figure 1 and 2, frequency of mammography for men should be included. In addition, labels of the figure axis should be added in all the bar graphs. 

  6. In line 317-328, there is an inconsistency in either the calendar time or the mean dose is in the parenthesis when providing the data. For example, “Mean dose received by women living in the least deprived area increased a 51% between 2007-2013 (0.57 mSv) and 0.86 mSv (2020-2021)”.

  7. In Line 363, reference should be added for the statement: “In addition, they did not take into account the need to consider radiation exposure when ordering imaging tests and the requirement to inform the patient about the risks associated with medical radiation exposure.”

  8. In Line 377, does modern helix CT cause a similar radiation dose compared to conventional CT? Does it cause inaccuracy in the mean dose calculation? If so, is it possible to recalculate the mean dose by classifying CT tests further into conventional CT and modern helix CT and use separate estimated radiation dose values?

Author Response

Response to Reviewer 1 Comments

The authors presented a retrospective study on the impact of recommendations to control exposure to radiation on diagnostic imaging frequency and dose with gender, age and socioeconomic status as factors. The study is properly designed and the conclusion revealed that there is a need to further improve the awareness and implementation of the recommendations and to pay additional attention regarding justification of diagnostic imaging in clinical practice. Overall, the article is organized and well-written. I would suggest publishing after the following comments are addressed.

  1. In Line 33, the last sentence in the abstract is not finished.

Done

  1. In Line 40-42, reference should be added for the statement: “Over the last 22 years, radiation due to CT exposures were estimated to account for 0.7% of the cancer incidence and 1% of cancer mortality”.

The reference is the number 4, which also refers to the sentence: ‘Moreover, the incidence of cancer in individuals who had been exposed to CT was found to be 24% higher compared to individuals who had not’. We have added the number 4 to clarify this point (introduction, page 1, line 44).

  1. In Line 173, can the author check on the minimum deprivation index value? Is it -284 or -2.84? In addition, details on the reconstruction method of deprivation index and the deprivation index ranges for the three deprivation levels should be provided in the article.

It is a mistake. The correct value is -2.84.

In addition, we have added details on the reconstruction method of the index in the method section (page 4, lines 207-222).

  1. There are multiple places in the article, where “,” and “.” are misused in numbers, making it difficult to comprehend, such as “750.9 tests per 1.000 men” in Line 204. The authors should go through the manuscript and fix all the misuse.

Thank you for your advice. We have checked the whole document to fix these mistakes.

  1. In Figure 1 and 2, frequency of mammography for men should be included. In addition, labels of the figure axis should be added in all the bar graphs. 

Done.

  1. In line 317-328, there is an inconsistency in either the calendar time or the mean dose is in the parenthesis when providing the data. For example, “Mean dose received by women living in the least deprived area increased a 51% between 2007-2013 (0.57 mSv) and 0.86 mSv (2020-2021)”.

We corrected the mistake (result section, page 13, lines 705-715).

  1. In Line 363, reference should be added for the statement: “In addition, they did not take into account the need to consider radiation exposure when ordering imaging tests and the requirement to inform the patient about the risks associated with medical radiation exposure.”

This affirmation also belongs to the reference 36. We have also added the number 36 at the end of this sentence to clarify the point (page 15, line 953).

  1. In Line 377, does modern helix CT cause a similar radiation dose compared to conventional CT? Does it cause inaccuracy in the mean dose calculation? If so, is it possible to recalculate the mean dose by classifying CT tests further into conventional CT and modern helix CT and use separate estimated radiation dose values?

We got the data from the Medical Image Bank of the Valencian Community from the Department of Universal Health and Public Health Service and did not distinguish between the slice-spiral CT and the multi-slice CT. However, according to a previous study (Brix G, Nagel HD, Stamm G, Veit R, Lechel U, Griebel J, Galanski M. Radiation exposure in multi-slice versus single-slice spiral CT: results of a nationwide survey. Eur Radiol, 2003, 8, 1979-91), the average effective dose to patients was only lightly changed from 7.4 mSv at single-slice to 8.1 mSv at quad-slice scanners, respectively. In our study, we applied an estimation of the average values of the effective dose of imaging tests, and thus, we do not consider that this variation according to the type of CT could have influenced on the results.
We have included this description in the limitation section (page 16, lines 1032-1040).

Author Response

Response to Reviewer 2 Comments

Title:

Good title, but it’s too long try to deduct the title no need for full details.

Following the reviewer’s advice, we have reduced the title.

Abstract,

Try to give direct and clear conclusion in your abstract.

We have modified the conclusion of the abstract.

Study design:

……performed in routine practice…… what do you by routine practice in other word, what are the practices that you have excluded?

With ‘routine practice’, we refer to the imaging tests carried out during the clinical practice. To clarify the expression, we have changed ‘routine practice’ by ‘clinical practice’ (method section, page 3, line 159). As we described in the method section, we included CT, radiography (mammography, conventional radiography and fluoroscopy) and nuclear medicine. We excluded imaging tests that did not involve radiation exposure (i.e., MRI and ultrasound) (page 3, lines 170-175).

Participants

What about nuclear medicine imaging?

Following the reviewer’s advice, we have included nuclear medicine diagnostic procedures.

2.5 Effective dose estimate

We based our estimates on ICRP103, except in those cases where we did not have enough information. This is not clear! Do you mean that patients that have no enough information were not estimated based on ICRP103? If that, so how you did the estimations for those patients?

The reviewer is right. That sentence was a mistake and we deleted it. All the effective doses were estimated based on ICRP103 (pages 3-4, lines 191-205).

Socioeconomical status

Your categorizations performed based on SEE!? Please try to give more clarification here?

We have included a more detailed description of the calculation of the deprivation index in the method section (page 4, lines 207-222).

Results

Distribution of demographic variables was similar between men and women across the period of study…what do you mean by similar??

We refer to the fact that there were not statistically differences according to the distribution of demographic variables (age and deprivation level) between men and women in the years 2007-2021. We have modified the sentence to clarify the point (result section, pages 5-6, lines 242-255).

According to your results, there is no mammography for the age period less than 18 years in all study years (calendar time)?!

Yes, there were mammographies for patients less than 18 years. However, the frequency was so low that we decided not to include them in the graphs. The calendar time has no effect in this frequency since we are evaluating a cross sectional study in which we do not follow the same patients along the period of study.

Page 10 line 308, In men, the mean dose associated to radiographs increased between 2007-2013 and 2014-2019 (0.14 mSv and 0.20, respectively), and decreased in 2020-2021 (0.17 mSv). How the mean dose reduced, and the number of radiographs showed in figure 1.1.A either for men or women were increases in the calendar tie 2014-2019??

According to table 1, the number of radiographs from men increased between 2007-2013 (440.5 per 1,000 men) and 2014-2019 (407.6 per 1,000 men) and it decreased in 2020-2021 (330.4 per 1,000 men). Thus, the mean dose followed a similar pattern.

In the figure 1, we can see as the number of radiographs increased in 2014-2019 only in subjects older than 64 years (in men and women) but it decreased in subjects <18 years and in those between 18-64 years.

Discussion

Page 13 line 406, However, it would be important to explore the pertinence of the imaging tests solicited and ensure that this reduction in imaging among the most deprived population, is not due to a reduction of the pertinent tests due to problems with access, as this could lead to diagnostic delays and other negative health outcomes.

This is not clear, and what do you mean other negative health outcomes??

We refer to diagnostic delays and the subsequent consequences of these delays (treatment, survival, etc). However, to clarify the point, we have deleted the expression ‘and other negative health outcomes’ (page 16, line 1017).

Page 13 line 410, We included a general hospital and its catchment are…. This statement is not clear??

It is a mistake. We added the letter ‘a’, so we can read, ‘We included a general hospital and its catchment area’ (page 16, line 1018).

Reviewer 3 Report

Congratulations for the excellent work , it  is well designed and well documented

Author Response

Thank you very much.